# The Impact of a De Facto CEO on Environmental, Social, and Governance Activities and Firm Value: Evidence from Korea

**Kil-Joo Baek and Young-Jun Yeo ***

Department of Accounting, Jeju National University, Jeju-si 63644, Republic of Korea; idakang@jdcenter.com
* Correspondence: yjyeo@jejunu.ac.kr; Tel.: +82-010-3900-0509

**Abstract:** This study analyzes the influence of CEO types on corporate governance, focusing on de facto (substantial) CEOs. We examine how substantial CEOs impact environmental, social, and governance (ESG) activities (Hypothesis 1) and corporate value (Hypothesis 2). Data were collected from KIS-VALUE and DART (Electronic Disclosure System) from the Financial Supervisory Service, defining substantial CEOs as the highest remuneration recipients who exceed the pay of the company's representative director. The results support Hypothesis 1, showing that companies with substantial CEOs are more likely to engage in ESG activities, potentially to improve public image while concealing self-serving behaviors. Hypothesis 2 is validated, indicating lower corporate value in companies with substantial CEOs, owing to the prioritization of personal interests over long-term profit maximization. Despite the limitations of exploring governance relationships beyond remuneration data, this study offers key contributions. It expands the research on corporate governance and ESG activities by identifying substantial CEOs through objective remuneration data. Additionally, it highlights the importance of an independent board for transparent governance and positive corporate value. Lastly, the empirical evidence shows the negative impact of misdirected ESG activities on corporate value. Using remuneration as an indicator, this study illuminates substantial CEOs' influences on corporate value and ESG activities, providing insights for future research in this area.

**Keywords:** corporate governance structure; de facto CEO (substantial CEO); corporate value; ESG activities

## 1. Introduction

### 1.1. Research Background

South Korea has a unique governance structure known as the chaebol. Scholars who view the chaebol governance structure positively have argued that when controlling shareholders participate directly in management as chief executive officers (CEO), they can address the issues of low economic growth rates through strong leadership, thereby exerting a positive influence on corporate value [1–3]. Conversely, scholars who view the chaebol governance structure negatively argue that when controlling shareholders and a small number of special-interest parties participate in management to pursue private benefits, their policy decisions can negatively affect corporate value [4,5].

Prior research has reported that the type of CEO is important as it has varying impacts on corporate value. Hong and Yoo [6] analyzed whether corporate performance is influenced by the type of CEO. They found that when the CEO is a professional manager, there is a statistically significant positive correlation with corporate performance. Moreover, they reported that this positive effect is amplified in companies that have independent and transparent governance structures. Additionally, negative effects are mitigated in firms that are controlled by dominant shareholders.

Ahn and Seo [7] were the first to define those shareholders who, within the owner-managers, evade legal responsibility while exercising influence on the company as "de facto CEOs", measured using compensation data. The measurement of CEO influence is inspired by the CPS (CEO pay slice) proposed by Bebchuk et al. [8], utilizing compensation data for

measurement. The CPS denotes the ratio of the CEO's salary to the total compensation of the top five executives within a company. Since Bebchuk et al. [8] first presented CPS as a metric to measure the influence of CEOs on corporate control and analyzed its relation to corporate value, it has been widely used.

In addition to this study, various other research has indicated that the impact of the CEO type on corporate value can vary based on factors such as leadership style, expertise, personality, compensation system, and corporate strategy.

In reality, there are cases where controlling shareholders assume the CEO role. However, there are also cases where they do not take on the CEO role, yet they still influence critical policy decisions within the company. An and Suh [7] used compensation data to objectively measure the authority and responsibility of employees in a company to identify the CEO. Then, they analyzed the impact of a top earner receiving more compensation than the CEO on the quality of a company's executive compensation disclosure. The analysis revealed that the quality of executive compensation disclosures is lower in companies with employees who receive more compensation than the CEO, suggesting that such companies may have an untransparent and vulnerable governance structure, warranting further analysis of its impact on corporate value.

My study followed the idea that, based on agency theory, there is a negative correlation with corporate value due to the agency costs of goal incongruity and resource wastage in companies where professional managers represent shareholders and dominant shareholders exercise substantial influence on corporate control with minority interests.

However, interest in corporate sustainability management has recently increased. Corporate sustainability is perceived as managing environmental, social, and economic risks while growing a business, thereby enhancing shareholder value. Corporations are engaging in environmental, social, and governance (ESG) activities to realize sustainable management, leading to a rising interest in ESG. Figure 1 illustrates the frequency of searches for the term "ESG" from 2004 to 2021 on the Google Trends analysis site, demonstrating a sharp increase starting in 2018.

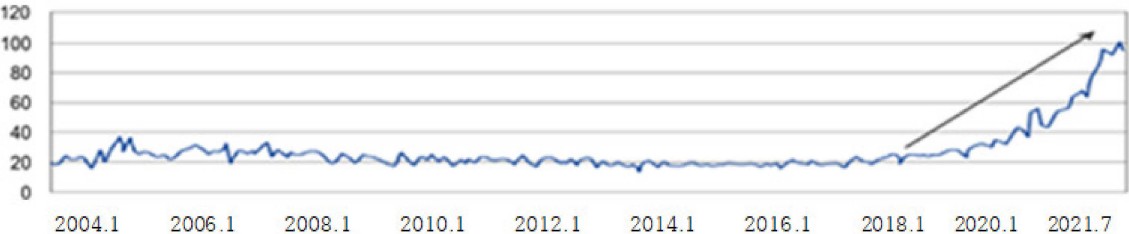

**Figure 1.** Trends in ESG Interest. Source: Google Trends Analysis Site [9].

However, there is considerable debate within academia about whether the amount spent on ESG activities should be viewed as an investment or a cost. Scholars who approach ESG expenditures as an investment concept often associate them with corporate innovation. On the other hand, scholars who view expenses on ESG activities as a cost often describe these activities as a means to improve the negative image of controlling shareholders and the company and to form a favorable public opinion to defend management rights. Although the impact of ESG activities on corporate value varies slightly among scholars, many studies universally accept that ESG activities are necessary for a company's sustainable management. As ESG activities have increased, studies have been actively conducted investigating the factors that influence corporate ESG activities. These studies empirically analyze how ESG affects topics related to accounting research, such as corporate value, earnings management, and financial performance. Various determinants appear to influence ESG activity. Considering that corporate governance is a key variable in determining ESG activities, it may vary depending on the CEO type. Thus, examining the impact of the presence or absence of CEOs in companies that exercise authority while avoiding legal responsibility for ESG activities is necessary.

*1.2. Research Objective*

This study analyzes the impact of the presence or absence of a substantial CEO on ESG activities and corporate value. In reality, while there are cases in which controlling shareholders directly participate in company management or delegate it to professional managers, there are also controlling shareholders who, under the nominal CEO, seek authority while avoiding legal responsibility. Samsung Electronics is a representative company in South Korea. Apart from Samsung Electronics, the controlling shareholders of socially criticized companies are not registered as CEOs who bear legal responsibilities; however, they exercise substantial influence through boards of directors and determine essential company policies.

Previous studies have asserted that firms with weak corporate governance can experience agency costs between professional and shareholder managers, thereby negatively influencing company value. Therefore, if a company pursues an increase in corporate value through profit maximization, the number of firms with de facto CEOs and weak corporate governance should decrease. However, as shown in Table 1, by examining the congruence between the CEO and the highest-paid employee in business reports from 2013 to 2021, we observe that the number of companies where employees are paid more than the CEO increases yearly.

**Table 1.** Proportion of companies with de facto CEOs by year. (Unit: Count, %).

|  | 2013 | 2014 | 2015 | 2016 | 2017 | 2018 | 2019 | 2020 | 2021 | SUM |
|---|---|---|---|---|---|---|---|---|---|---|
| CC * | 54 | 58 | 49 | 68 | 63 | 74 | 83 | 81 | 84 | 614 |
| Ratio | 8.8% | 9.4% | 8.0% | 11.1% | 10.3% | 12.1% | 13.5% | 13.2% | 13.7% | 100.0% |

* Company Count. Source: Data processed by the researcher based on Financial Supervisory Service data.

Studies assert that companies with top executives are prone to weak corporate governance and incur agency costs. An and Suh [7] identified the presence of executives who evaded legal responsibility while exercising power using compensation data and analyzed the impact of the highest-paid employee, who receives more compensation than the CEO, on the disclosure quality of corporate executive compensation. Their analysis suggests that de facto CEOs vigorously pursue personal benefits, asserting a negative correlation between this and disclosure quality.

In companies with de facto CEOs, corporate governance tends to be opaque and fragile, increasing the likelihood that crucial policies will be determined by a minority who will prioritize personal interests over corporate profits. In cases where a de facto CEO with self-interest exists, they may exploit ESG activities to create favorable public opinion, such as to protect managerial rights. A de facto CEO may be proactive in ESG expenditure to gain favorable public opinion, even if the company's financial performance declines. Recent studies have shown that environmental management (E), social responsibility (S), governance improvements (G), economic uncertainty, and volatility in operating profits can influence ESG activities [10,11]. Corporate governance significantly affects a company's performance and sustainability. Therefore, this study empirically analyzes the influence of a de facto CEO as a determining factor in ESG activities. Moreover, conflict can occur between the CEO and the de facto CEO, and the cost of resolving this conflict can negatively affect a company's value. Thus, we empirically analyze the influence of a de facto CEO on corporate value.

The remainder of this paper is organized as follows: In Section 1, the Introduction, we explain the background and purpose of this research. Section 2 examines previous studies on de facto CEO and ESG activities. In Section 3, we set up two research hypotheses and present a research model to test them, explaining the variables of the research model, data collection methods, and sample selection methods. In Section 4, we conduct descriptive statistical analysis, difference analysis between groups of interest variables, and correlation

analysis, and then explain the empirical analysis results of the hypotheses. Finally, Section 5 summarizes the study and discusses its contributions and limitations.

## 2. Literature Review

### 2.1. A De Facto CEO

Studies related to corporate governance transparency use the proportion of outside directors on the board and the presence of an independent audit committee as proxies for analyzing transparency. The results demonstrate that a higher proportion of outside directors on a company's board and the presence of an independent audit committee enable the company's decision-making process to be more transparent and independent. There is consistent evidence that such companies, being proactive in investor protection, experience increased investments from investors, leading to increased corporate value [1,12]. Giroud and Mueller [12] demonstrated that, under the assumption of inefficient markets, U.S. firms with better corporate governance have higher corporate value than those with weaker governance. Black et al. [1] argued that even when effectively controlling for endogeneity issues using 2SLS (two-stage least squares), firms with good corporate governance still exhibit higher corporate value than firms with weaker governance. From these studies, we can infer that as the transparency of a company's corporate governance increases, its corporate value also rises.

Research on chaebols (i.e., large business conglomerates) has revealed that their presence or absence has mixed effects on corporate value. Studies that argue against the existence of large business conglomerates have empirically shown that firms belonging to such conglomerates exhibit relatively lower financial efficiency [1,4] and corporate value than firms not affiliated with large business conglomerates [3]. In contrast, studies that argue for the positive impact of large business conglomerates on corporate value claim that government tax incentives and various forms of support for these conglomerates increase corporate value [13].

In the 2000s, there was a growing interest in understanding how the influence and characteristics of CEOs impact a company's financial performance and corporate value [6,14]. Bebchuk et al. [8] investigated the effect of CEOs on corporate financial performance and value; they introduced the CPS measure to gauge a CEO's influence on corporate governance and analyze its relationship with corporate value. The analysis reveals a negative correlation between CEO influence and corporate value. This finding supports traditional agency theory, suggesting that when the values pursued by managers and shareholders differ, CEOs may exercise their authority to maximize their interests, negatively affecting corporate value.

Studies have examined the relationship between de facto CEOs and various aspects of corporate governance, such as the disclosure quality of executive compensation, CEO overcompensation, human resource investment in internal accounting control systems, and cost stickiness. Particularly, research on the disclosure quality of executive compensation, CEO overcompensation, and human resource investment in internal accounting control systems analyzes how the board of directors' independence and expertise play a role when a de facto CEO is present. These studies can be regarded as related to corporate governance.

In many studies on multiple CEOs, the representative director in a business report is defined as the CEO. However, An and Suh [7] investigated the top ten companies with two or more executives receiving annual compensation of over KRW 500 million. Their findings revealed that the highest-paid executives were the dominant shareholders in most of these top companies, while the second-highest-paid executives were professional managers. Building on this insight, this study highlights the likelihood of a de facto CEO's existence within a company.

Based on the executive compensation data disclosed since 2013, we define the top earner who receives more compensation than the CEO as the de facto CEO. By defining a company's de facto CEO based on compensation data, we can identify employees or executives who exert the most significant influence, regardless of their official title. Additionally,

this approach allows us to verify and identify the real CEO, even in companies where the representative director is not specified in the business report or where there are multiple co-CEOs. This method effectively identifies the individual with the highest influence and authority within the company, regardless of their position. It enables the validation of companies whose CEO's identity may not be explicitly indicated in the business report or those with joint representation by multiple CEOs.

The analysis [7] of the relationship between a de facto CEO and executive compensation disclosure quality revealed that companies with a de facto CEO tend to have lower executive compensation disclosure quality than companies without a de facto CEO. A de facto CEO's pursuit of private interests significantly contributes to this finding.

This study [7] examines the relationship between a de facto CEO and excess compensation and analyzes whether an independent and professionally diverse board of directors can effectively constrain a de facto CEO. This analysis provides evidence that companies with de facto CEOs exhibit significantly higher excess compensation for their top earners. Moreover, it was demonstrated that a higher proportion of outside directors, separation of CEO and board chair roles, and a higher level of expertise among board members could effectively restrain excess compensation for a de facto CEO. In particular, the presence of professionally diverse outside directors on a company's board of directors showed a stronger inhibitory effect on excess compensation for a de facto CEO than for companies without such expertise. Based on these findings, it was argued that to enhance the effectiveness of a company's board of directors, it should operate independently to provide appropriate advice and practical constraints on the CEO's management policies; additionally, including outside directors with expertise in various fields is essential.

This study [7] examines how the presence of a de facto CEO affects human resource investment in internal accounting control systems and further analyzes whether the independence and expertise of the board of directors also influence this relationship. The analysis [7] reveals that companies with de facto CEOs tend to have a lower ratio of internal accounting personnel than companies without de facto CEOs. Additionally, the effect of a de facto CEO on human resource investment in internal accounting control systems weakens as the board of directors' independence increases.

Ko and Jung [15] conducted an empirical analysis of whether the presence of a de facto CEO influences the cost stickiness of selling expenses after the COVID-19 pandemic. The analysis revealed that companies with a de facto CEO exhibited alleviated downward cost stickiness compared to companies without a de facto CEO after the COVID-19 pandemic.

### 2.2. ESG Activities

ESG activities, representative decision factors for sustainable management, have been actively researched in various fields such as accounting, financial management, business strategy, and marketing. In accounting studies, research has been conducted on various topics such as accounting transparency, corporate value, financial performance, and earnings management. Prior research on corporate value has shown mixed results.

Studies claiming a negative impact of ESG on corporate value are grounded in the theory of agency problems. Dominant shareholders tend to waste company resources to pursue their private benefits, and while professional managers aim to maximize shareholder value, controlling shareholders might be oriented toward the maximization of their private gains, leading to a potential misalignment of goals [16–18]. Based on the theory of agency problems, prior research analyzing the impact of ESG on corporate value has reported a negative association between ESG activities and corporate value [19–22].

Barnett and Salomon [19] argued that when a corporation pursues ESG activities, it is difficult to seek financial performance and social benefits simultaneously. Therefore, the relationship between ESG activities and financial performance is not dichotomous but exhibits a curvilinear relationship. Surroca et al. [20] recognized ESG activities as equivalent to a corporation's intangible resources such as reputation, brand value, and social networks, and conducted an empirical analysis of their impact on financial performance.

Additionally, Fisher-Vanden and Thorburn [21] proved that the value of corporations intensifying environmental management decreases due to negative cumulative abnormal returns. Lyon et al. [22] indicated that in the Chinese market, the corporate value of companies that are recognized for their environmental management and have won awards shows no significant difference or even a decrease compared to those that are not recognized.

## 3. Research Methodology

### 3.1. Hypothesis Development

Numerous studies have been conducted on how an increase in ESG activities impacts accounting-related factors. However, research on the determinants of ESG activity is insufficient. Since 2011, the Korea ESG Standards Institute has announced annual ESG ratings. A corporation's ESG activities are determined by the influence of each component constituting ESG (environmental management, social responsibility, and governance) [11,23,24]. These ESG components are influenced by the CEO's level of ethical consciousness [23] and are affected by profitability indicators such as sales and operating profit. Research has shown that they decrease as economic uncertainty increases [24].

Unlike owner-managers, de facto CEOs informally influence a company; thus, they can receive a significant amount of scrutiny and criticism from civic groups and the media. If criticism becomes widespread, they may lose influence. Therefore, there are incentives for de facto CEOs to actively participate in ESG activities, which are non-profit activities, for various reasons, such as forming favorable public opinion about themselves and strengthening their control over the company. De facto CEOs are likely to prefer increasing ESG-related expenditures rather than reducing them because using ESG to maximize their benefits and reputation is far more advantageous. Based on this fact, we hypothesized (Hypothesis 1) that the presence of a de facto CEO positively impacts ESG activities.

**Hypothesis 1:** *Companies with a de facto CEO are more proactive in ESG activities than those without one.*

Studies have asserted that South Korea's rapid economic growth is possible because of the unique corporate structure of chaebols. However, the empirical results from most studies have reported that chaebol corporations, owing to their vulnerable governance structures, negatively influence corporate value [3,4,12,25]. Specifically, research has found that the corporate value of chaebol firms is lower than that of firms that do not belong to chaebols [3], and it has been proven that the financial efficiency of chaebol firms is relatively low [4]. Additionally, research has argued that stakeholders such as shareholders and managers, shareholders and bondholders, controlling shareholders and minority shareholders, management, and employees may have differing interests, leading to potential conflicts in the decision-making process, which can subsequently negatively impact corporate value [26].

This study intends to determine the process of identifying the CEO of a company using compensation data; these have been employed in prior studies to assess the CEO's influence and analyze the impact on corporate value from a governance perspective. The company's CEO should be the highest-paid individual with the greatest authority and responsibility. Various internal, external, legal, and institutional checks and balances exist within a corporation, making compensation one of the fairest measures to objectively assess the authority and responsibilities of employees and executives. CEOs representing ordinary shareholders' interests have incentives to maximize their corporate value and extend their tenure. However, if a de facto CEO exists within a corporation, they may have incentives to make decisions that enhance their reputation or pursue personal benefits rather than activities that would benefit the company. Thus, if conflicts arise between the formal CEO and de facto CEO during the corporate policy decision-making process, the company can incur agency costs to resolve these conflicts, potentially negatively impacting corporate value. An and Suh [7] argued that if a registered or non-registered executive receives higher

compensation than the CEO, the top compensated individual in the corporation could wield greater influence over important corporate policy decisions than the CEO through means such as the board of directors.

The direct influence of ESG activities on corporate value can be easily confirmed by reviewing previous studies. However, satisfying the objectives of both non-profit activities for societal benefits, such as ESG, and profit-driven activities for financial benefits, is not easy for corporations. As such, investing in ESG activities may entail considerable short-term expenditures, which could negatively affect financial performance [11,27–29]. Concentration on non-profit ESG activities may distract the core business, potentially lowering corporate competitiveness and resulting in a decline in corporate value [30,31]. Barnea and Rubin [31] argued that a CEO's ESG activities can enhance individual rather than corporate reputation, thus negatively impacting corporate value. Based on these facts, we set Hypothesis 2, expecting that the highest-paid individuals receiving more compensation than the CEO in the business report negatively affect corporate value.

**Hypothesis 2:** *The presence of a de facto CEO results in lower corporate value compared to its absence.*

*3.2. Methodology*

For the research model of Hypothesis 1, the dependent variable, ESG, was measured based on the criteria provided by the Korea ESG Standards Institute. The ESG rating was coded as 1 if it was B+ or higher and 0 otherwise. The independent variable, RealCEO, was defined as a dummy variable, coded 1 if the individual with the highest salary in the company was not the representative director and 0 if they were. Control variables that could influence ESG included firm size (SIZE), debt ratio (LEV), return on equity (ROE), sales growth rate (SG), affiliation with a major corporate group (CH), foreign ownership ratio (Foreign), and year and industry dummy variables.

Larger firms, which typically have lower levels of unsystematic risk and a stronger financial structure, have greater potential to focus not only on their core profit-making business but also on non-profit activities like ESG. Thus, firm size (SIZE) was included as a control variable, measured by taking the natural logarithm of total assets [20]. Firms with a high debt ratio (LEV) may have an incentive to actively pursue ESG activities to present a favorable image to investors and stakeholders; hence, LEV was included in the model and was measured by dividing debt by equity [32].

Companies with higher profitability are likely to invest more resources in ESG activities. Therefore, ROE was included as a control variable and computed by dividing net income by total equity [33].

The sales growth rate (SG) reflects a company's growth prospects; it is anticipated that with higher future profitability, more resources can be allocated to ESG activities. Thus, SG was included as a control variable, and it was measured by subtracting the previous year's sales from this year's sales and then dividing it by last year's sales [34]. As major corporate groups are likely to be more actively engaged in socially responsible activities, the CH variable, indicating whether a company is part of the top 30 chaebols in South Korea as per the Fair Trade Commission's statistics of May 2022, was included in the control variables. It was coded 1 if the company was part of the chaebols, and 0 otherwise [13]. A higher foreign ownership ratio (foreign) suggests that a company might be required to adhere to stricter global ESG standards. Therefore, it was included as a control variable and measured using the foreign ownership ratio provided by KIS-Value [35]. The model also incorporated year and industry dummy variables to control for yearly and industry-specific effects.

For the research model of Hypothesis 2, Tobin's Q was used as a proxy for firm value and computed by dividing the sum of the market value of equity and the book value of debt by total assets. Control variables that could affect firm value included ESG activities (ESG), firm size (SIZE), debt ratio (LEV), return on equity (ROE), sales growth rate (SG),

affiliation with a major corporate group (CH), ownership stake of the largest controlling shareholder (Ownerrate), and year and industry dummy variables.

The inclusion of firm size (SIZE) and debt ratio (LEV) was justified by their potential to influence the risk related to firm value. Specifically, large firms and those with a low debt ratio face a reduced risk of bankruptcy. ROE was included to control for the influence of profitability, and SG was incorporated due to its potential effect on firm value. Companies with a higher ratio of cash included in earnings are generally considered to have higher-quality earnings; hence, the operating cash flow ratio (CFO) was also included [36,37]. A higher ownership stake (Ownerrate) by the largest controlling shareholder can expedite major investment decisions, such as those related to facility investments, and enhance firm value. However, it also carries the risk of unilateral decision-making, which can negatively impact firm value. Hence, it was included as a control variable, measured by the stake of the largest controlling shareholder [38]. From a governance perspective, the CH variable, indicating affiliation with a major corporate group, was included as a control variable [39]. Yearly and industry-specific influences were controlled for by incorporating annual and industry dummy variables into the model.

$$ESG_{it} = \alpha_0 + \alpha_1 RealCEO_{it} + \alpha_2 SIZE_{it} + \alpha_3 LEV_{it} + \alpha_4 ROE_{it} + \alpha_5 SG_{it} + \alpha_6 CH_{it} + \alpha_7 Foreign_{it} + \alpha_8 \Sigma ICODE_{it} + \alpha_9 \Sigma YR_{it} + \varepsilon_{it} \tag{1}$$

$$Tobin's Q_{it} = \alpha_0 + \alpha_1 RealCEO_{it} + \alpha_2 ESG_{it} + \alpha_3 SIZE_{it} + \alpha_4 CFO_{it} + \alpha_5 LEV_{it} + \alpha_6 ROE_{it} + \alpha_7 SG_{it} + \alpha_8 CH_{it} + \alpha_9 Ownerrate_{it} + \alpha_{10} \Sigma ICODE_{it} + \alpha_{11} \Sigma YR_{it} + \varepsilon_{it} \tag{2}$$

*3.3. Variables*

3.3.1. A De Facto CEO

Finkelstein et al. [40] defined the CEO as the individual who leads the overall performance and decision-making of the organization and bears ultimate responsibility. The concept of "a de facto CEO", the main variable of interest in this study, stems from the premise that employee compensation in a firm is an objective indicator that assesses employee abilities and is proportionally determined according to the extent of employees' impact on the company. The purpose of a company is to maximize profits for its continued survival; thus, it is expected that the highest compensation should be paid to the CEO, who holds legal responsibility and decision-making authority in company operations. However, if an employee receives more compensation than the CEO, it can be inferred that the individual exerts more influence on the company's policy decisions.

The "Capital Market and Financial Investment Business Act" and its enforcement regulations, which mandate the disclosure of individual executive compensation of KRW 500 million or more in business reports, were enacted on 27 August 2013. Consequently, publicly traded companies on the Korea Stock Exchange and KOSDAQ began disclosing executive compensations of KRW 500 million or more in their business reports. The compensation and positions of registered and nonregistered executives, which are used to determine the existence of a real CEO, can be verified using the data disclosed in business reports.

This study adopts the measurement method of a de facto CEO, used by An and Suh [7], defining the highest-remunerated individual as a de facto CEO when their remuneration exceeds that of the representative director, as per the business report. Specifically, the company's compensation data were used to measure the de facto CEO variable. Compensation data were collected via the online open API provided by the Financial Supervisory Service's electronic disclosure system (dart.fss.or.kr) for individual compensation data, coupled with the manual collection of the names of the representative directors disclosed in the business reports. A dummy variable was defined for a de facto CEO: if the highest compensated individual in the company was the representative director, the value was 0; otherwise, it was 1.

3.3.2. ESG Activities

ESG activities are defined as the actions taken by corporations in the pursuit of sustainable management. Through ESG activities, corporations can contribute to creating environmental, social, and economic value and positively impact stakeholders, such as investors.

This study measured ESG activities using ESG ratings provided by the Korea ESG Standards Institute, which are most commonly used in South Korean research. The ESG evaluation model of the Korea ESG Standards Institute is designed to faithfully reflect not only international standards such as the OECD Corporate Governance Principles and ISO26000 but also domestic laws and business environments. This model comprises five stages: (1) collecting basic data; (2) conducting preliminary and in-depth evaluations; (3) verifying evaluation results; (4) sharing ESG evaluation results with target companies and implementing bidirectional feedback; and (5) finalizing and disclosing ESG evaluation grades. Specifically, the model utilizes corporate disclosure materials, institutional data (such as supervisory agencies and local governments), and media materials to collect over 900 basic data for approximately 900 listed companies. The basic data were categorized into 18 major classifications and 265 key evaluation items, after which a preliminary evaluation was conducted. An in-depth evaluation was then performed using 58 key evaluation items to determine whether ESG-related issues could impair corporate value. The completed evaluation results were verified, and bilateral feedback was provided to the evaluated companies through a web-based evaluation system, enhancing the reliability of the evaluation results. Based on the derived scores, the ESG Committee of the Korea ESG Standards Institute assigns grades. If an event that could affect the rating occurs after the evaluation period, the company is classified as a rating adjustment review candidate and reexamined. Additionally, they select excellent companies and disclose grades of B+ or higher in the overall market, corporate governance, and environmental and social sectors. Since 2011, the Korea ESG Standards Institute has announced annual ESG ratings, including social responsibility, governance, and environmental management. Therefore, in this study, we intended to measure ESG activities using a dummy variable that assigned 1 to companies with a comprehensive ESG rating of B+ or higher and 0 otherwise, based on the annual ESG ratings of approximately 900 companies announced by the Korea ESG Standards Institute.

*3.4. Data*

The data on the highest executives were collected using the Financial Supervisory Service's Dart Electronic Disclosure System (dart.fss.or.kr, accessed on 1 February 2022). Aligned with the amendments to Article 159 (2) of the Capital Market Act of 2013, companies were required to disclose the top five registered executives who received over KRW 500 million. Further amendments in 2018 mandated the disclosure of the remuneration status of the top five employees and executives, regardless of their registration status. Consequently, data were collected from the start of these disclosures, from 2013 to 2021. The names of the representative directors listed in the business report and information on the top five earners in a company were manually collected. General company information and original disclosure documents provided by the Dart Electronic Disclosure System were received in XML format from 2013 to 2021. Data, including the names of the representative directors in the business report, the overall remuneration status of the directors and auditors (amount approved by the shareholders' meeting), executive status, individual remuneration status of the directors and auditors, and individual remuneration payments (the top five exceeding KRW 500 million), which began to be disclosed in 2018, were converted to Excel data for collection. The total compensation for individual executives included employment income, other income, and retirement income, of which one-time income, such as other income and retirement income, was excluded from the sample.

The financial data for each company were obtained by downloading the massive amount of data provided by Kis-Value. The KOSDAQ market, which has significantly fewer individual executive compensation disclosures than the Korea Exchange (KOSPI) market, was excluded to ensure homogeneity with previous studies. Data were collected from

companies listed on the KOSPI as of December 2021. Firms that continuously possessed financial information from 2013 to 2021, had a December fiscal year-end, were not insolvent, and had not operated at a deficit for three consecutive years were selected as samples. To ensure homogeneity among the samples, the insurance and finance industries were excluded according to the industrial classification, and industries with fewer than ten firms per year based on the medium classification used in the Korean Standard Industrial Classification were also excluded to finalize the sample. The detailed sample selection process is presented in Table 2. To control for the influence of outliers, winsorization was performed on the top and bottom 1% of the samples for the dependent and control variables, which were continuous. Table 2 shows the collected sample before the empirical analysis.

**Table 2.** Detailed sample selection process.

| Sample Classification | Observations |
|---|---|
| Companies listed on the stock market with a December fiscal year-end | 7020 |
| Companies without financial data continuously from 2013 to 2021 | (1389) |
| Companies with capital erosion or operating losses for three consecutive years | (1841) |
| Companies without ESG ratings or information on the de facto CEO | (1421) |
| Industries with less than ten categories based on the medium classification of the Korean Standard Industrial Classification | (69) |
| Final companies for analysis | 2300 |

Table 3 shows the annual proportion of de facto CEO between 2013 and 2021. The figure shows the yearly percentage of companies with a de facto CEO out of the 2300 sample companies, with 614 companies having a de facto CEO.

**Table 3.** Companies listed on the stock market each year with a substantial top executive.

| Year | Total | | a De Facto CEO | | Others | |
|---|---|---|---|---|---|---|
| | Observations | Ratio | Observations | Ratio | Observations | Ratio |
| 2013 | 195 | 8.5% | 54 (8.8%) | 27.7% | 141 | 72.3% |
| 2014 | 210 | 9.1% | 58 (9.4%) | 27.6% | 152 | 72.4% |
| 2015 | 210 | 9.1% | 49 (8.0%) | 23.3% | 161 | 76.7% |
| 2016 | 242 | 10.5% | 68 (11.1%) | 28.1% | 174 | 71.9% |
| 2017 | 247 | 10.7% | 63 (10.3%) | 25.5% | 184 | 74.5% |
| 2018 | 286 | 12.5% | 74 (12.0%) | 25.9% | 212 | 74.1% |
| 2019 | 295 | 12.8% | 83 (13.5%) | 28.1% | 212 | 71.9% |
| 2020 | 311 | 13.5% | 81 (13.2%) | 26.0% | 230 | 74.0% |
| 2021 | 304 | 13.3% | 84 (13.7%) | 27.6% | 220 | 72.4% |
| SUM | 2300 | 100.0% | 614 (100.0%) | 26.7% | 1686 | 73.3% |

*3.5. Descriptive Statistics and Correlation Analysis*

Table 4 presents the descriptive statistics for the variables used in the empirical analysis.

Table 5 presents the t-test results conducted to ascertain if a statistical difference exists between companies with real CEOs (de facto CEOs) and those without them. Of the sample of 2300 companies, 614 (26.7%) had a RealCEO. The average Tobin's Q of companies with RealCEOs is 1.10, exhibiting a statistically significant 0.21 decrease at the 1% level, compared to the average of 1.32 for companies without RealCEOs. Therefore, there is a difference in corporate value between companies with and without real CEOs. Moreover, the differences between the groups can be confirmed at the 5% level for ESG activities (ESG)

and the maximum controlling shareholder stake (Ownerrate). The average large corporate group value for companies with RealCEOs was 0.22, surpassing the average of 0.17 by 0.05. However, the maximum shareholder stake was 0.02 lower, suggesting that large corporate groups might have a vulnerable governance structure where the majority shareholder can wield significant influence over a company.

**Table 4.** Descriptive statistics (N = 2300).

| Variable | Mean | Std. Dev. | Min | 1st Qnt. | Median | 3rd Qnt. | Max |
|---|---|---|---|---|---|---|---|
| Tobin's Q | 1.26 | 0.95 | 0.35 | 0.75 | 0.99 | 1.39 | 6.31 |
| RealCEO | 0.27 | 0.44 | 0.00 | 0.00 | 0.00 | 1.00 | 1.00 |
| ESG | 0.33 | 0.47 | 0.00 | 0.00 | 0.00 | 1.00 | 1.00 |
| SIZE | 26.58 | 1.54 | 23.98 | 25.45 | 26.27 | 27.45 | 31.18 |
| CFO | 0.06 | 0.05 | 0.00 | 0.03 | 0.05 | 0.09 | 0.24 |
| LEV | 0.88 | 1.03 | 0.01 | 0.22 | 0.52 | 1.15 | 5.91 |
| SG | 0.05 | 0.23 | −0.62 | −0.05 | 0.02 | 0.11 | 1.39 |
| ROE | 0.07 | 0.07 | 0.00 | 0.03 | 0.05 | 0.09 | 0.45 |
| Ownerrate | 0.29 | 0.16 | 0.03 | 0.17 | 0.25 | 0.39 | 0.86 |
| Foreign | 0.10 | 0.12 | 0.00 | 0.02 | 0.06 | 0.13 | 0.78 |
| CH | 0.19 | 0.39 | 0.00 | 0.00 | 0.00 | 0.00 | 1.00 |

**Table 5.** Analysis of group differences depending on the presence of a de facto CEO.

| Variable | RealCEO = 0 | | | RealCEO = 1 | | | *p*-Value |
|---|---|---|---|---|---|---|---|
| | Observations | Mean | Std. Dev. | Observations | Mean | Std. Dev. | |
| Tobin's Q | 1686 | 1.32 | 1.02 | 614 | 1.11 | 0.69 | 0.000 |
| ESG | 1686 | 0.32 | 0.47 | 614 | 0.36 | 0.48 | 0.049 |
| Size | 1686 | 26.61 | 1.54 | 614 | 26.51 | 1.52 | 0.182 |
| CFO | 1686 | 0.06 | 0.05 | 614 | 0.06 | 0.05 | 0.755 |
| LEV | 1686 | 0.89 | 1.06 | 614 | 0.85 | 0.93 | 0.491 |
| ROE | 1686 | 0.07 | 0.07 | 614 | 0.07 | 0.07 | 0.173 |
| SG | 1686 | 0.05 | 0.23 | 614 | 0.05 | 0.24 | 0.981 |
| CH | 1686 | 0.17 | 0.38 | 614 | 0.22 | 0.42 | 0.007 |
| Ownerrate | 1686 | 0.30 | 0.16 | 614 | 0.28 | 0.14 | 0.049 |
| Foreign | 1686 | 0.10 | 0.12 | 614 | 0.10 | 0.11 | 0.904 |

Tobin's Q: Measured as a proxy for firm value, this variable is calculated by dividing the sum of the market value of common and preferred stock and the book value of short-term and long-term debt by the total assets of the company.

ESG: This variable represents a firm's ESG (environmental, social, and governance) activities. It is measured using the ESG Index released annually by the Korea ESG Standards Institute. A dummy variable of 1 is assigned if the firm's ESG rating is B+ or higher, and 0 otherwise.

Size: This variable represents the size of the firm. To control for the size effect, the natural logarithm of the firm's total assets is used.

LEV: This variable represents the firm's leverage ratio and is calculated by dividing the firm's total liabilities by its total equity.

CFO: This variable represents the cash flow from operating activities ratio and is measured by dividing the firm's operating cash flows by its total assets.

ROE: This variable represents the return on equity and is calculated by dividing the firm's net income by its total equity.

SG: This variable represents the firm's sales growth rate, measured by subtracting last year's sales from this year's sales and dividing by last year's sales.

CH: This variable is unique to South Korea and represents whether the firm is among the top 30 conglomerates in South Korea based on asset size (including subsidiaries). It is measured as a dummy variable, with 1 indicating a conglomerate and 0 otherwise.

Ownerrate: This variable represents the ownership percentage of the firm's largest shareholder and is measured based on the share ownership rate of the firm's single largest shareholder.

Table 6 presents the correlations between the variables used in this study. Tobin's Q, the main variable of interest, had a statistically negative correlation with RealCEO at the 1% level. It showed a positive correlation at the 5% level with the ESG dummy variable, representing the ESG activities provided by the Korean ESG Standards Institute. Control variables—company size (Size), cash flow from operations (Cfo), return on equity (ROE), sales growth rate (SG), and maximum controlling shareholder stake (Ownerrate)—were found to have a statistically positive correlation at the 5% level. The debt ratio (LEV) was found to have a statistically negative correlation at the 5% level. However, no statistically significant results were found for the chaebol (CH) dummy variable, representing large corporate groups.

**Table 6.** Pearson correlation analysis.

| Variable | Tobin's Q | RealCEO | ESG | Size | CFO | LEV | ROE | SG | CH | Ownerrate | Foreign |
|---|---|---|---|---|---|---|---|---|---|---|---|
| Tobin's Q | 1 | | | | | | | | | | |
| RealCEO | −0.10 *** | 1 | | | | | | | | | |
| ESG | 0.05 ** | 0.04 ** | 1 | | | | | | | | |
| Size | 0.36 *** | −0.03 | 0.48 *** | 1 | | | | | | | |
| CFO | 0.22 *** | −0.01 | 0.06 *** | 0.17 *** | 1 | | | | | | |
| LEV | −0.28 *** | −0.01 | −0.03 | −0.25 *** | −0.09 *** | 1 | | | | | |
| ROE | 0.23 *** | −0.03 | 0.04 * | 0.13 *** | 0.45 *** | 0.00 | 1 | | | | |
| SG | 0.13 *** | 0.00 | 0.05 ** | 0.05 *** | 0.03 | −0.06 *** | 0.18 *** | 1 | | | |
| CH | −0.01 | 0.06 *** | 0.37 *** | 0.57 *** | 0.08 *** | 0.03 | 0.06 *** | 0.00 | 1 | | |
| Ownerrate | 0.06 *** | −0.04 ** | 0.02 | −0.02 | 0.03 | −0.04 * | 0.04 ** | 0.04 * | 0.00 | 1 | |
| Foreign | 0.12 *** | −0.00 | 0.29 *** | 0.49 *** | 0.24 *** | −0.12 *** | 0.14 *** | 0.01 | 0.32 *** | 0.00 | 1 |

Significance levels represented by *p*-values in the correlation analysis are as follows: *** represents a 1% significance level, ** represents a 5% significance level, and * represents a 10% significance level.

## 4. Empirical Analysis

### 4.1. The Influence of De Facto CEOs on ESG Activities

In the empirical analysis for Hypothesis 1, to test for endogeneity between the dependent variable ESG and the independent variable RealCEO, we utilized the proportion of treasury stocks owned by the company as an instrumental variable, as employed by Park and Lee [13]. They argued that while there exists a correlation between corporate governance structure and the proportion of treasury stocks—as firms with weaker governance structures might be incentivized to increase their proportion of treasury stocks as a means to defend and maintain their management rights—there is no direct influence on ESG. In this study, to ascertain the appropriateness of using the proportion of treasury stocks and lagged variables as instrumental variables, we conducted the J-test. The results did not reject the null hypothesis, confirming the suitability of these as instrumental variables. The empirical results of the 2SLS analysis in Table 7 were not significantly different from those obtained using ordinary least squares (OLS) regression, verifying that there were no endogeneity issues.

**Table 7.** Empirical analysis results of Hypothesis 1.

| Variable | Variable = ESG | | | |
| :---: | :---: | :---: | :---: | :---: |
| | **OLS** | | **2SLS** | |
| | **Coefficient** | **T-Value** | **Coefficient** | **T-Value** |
| Intercept | −2.95 | −13.63 | −2.95 | −13.57 |
| RealCEO | 0.05 *** | 2.59 | 0.06 ** | 2.01 |
| SIZE | 0.12 *** | 14.87 | 0.12 *** | 14.83 |
| LEV | 0.04 *** | 3.75 | 0.04 *** | 3.67 |
| SG | 0.08 ** | 2.11 | 0.08 ** | 2.14 |
| ROE | −0.20 | −1.61 | −0.21 * | −1.67 |
| CH | 0.14 *** | 4.92 | 0.15 *** | 5.00 |
| Foreign | 0.29 *** | 3.16 | 0.29 *** | 3.18 |
| Fixed Effects | Year and Industry | | Year and Industry | |
| Observations | 2300 | | 2300 | |
| F-statistics | 20.20 *** | | 20.12 *** | |
| J-test (*p*-value) | - | | 2.30(0.317) | |
| Adjusted R2 | 0.269 | | 0.268 | |

Significance levels represented by *p*-values in the correlation analysis are as follows: *** represents a 1% significance level, ** represents a 5% significance level, and * represents a 10% significance level.

In the empirical analysis for Hypothesis 2, to test for endogeneity between the dependent variable, firm value, and the independent variable, RealCEO, we utilized lagged variables as instrumental variables. Given that there was only one instrumental variable and one explanatory variable, the situation was considered as "just identified," hence the overidentification test was not conducted. Instead, we proceeded with the weak IV test. With an F-statistic value exceeding 10, the instrumental variable was deemed strong, confirming its appropriateness. The empirical results of the 2SLS analysis in Table 8 were not significantly different from those obtained using OLS, verifying that there were no endogeneity issues.

**Table 8.** Empirical analysis results of Hypothesis 2.

| Variable | Variable = Tobin's Q | | | | | | | |
| :---: | :---: | :---: | :---: | :---: | :---: | :---: | :---: | :---: |
| | **Z1 (OLS)** | | **Z2 (OLS)** | | **Z3 (OLS)** | | **Z4 (2SLS)** | |
| | **Coefficient** | **T-Value** | **Coefficient** | **T-Value** | **Coefficient** | **T-Value** | **Coefficient** | **T-Value** |
| Intercept | −6.20 | −15.92 | −7.15 | −17.62 | −6.99 | −17.14 | −6.92 | −16.87 |
| RealCEO | −0.14 *** | −3.62 | - | - | −0.12 *** | −3.30 | −0.19 *** | −3.42 |
| ESG | - | - | −0.25 *** | −6.23 | −0.24 *** | −6.05 | −0.24 *** | −6.09 |
| SIZE | 0.27 *** | 18.84 | 0.31 *** | 20.25 | 0.30 *** | 19.91 | 0.30 *** | 19.81 |
| CFO | 1.48 *** | 3.58 | 1.45 *** | 3.52 | 1.47 *** | 3.59 | 1.44 *** | 3.51 |
| LEV | −0.10 *** | −5.59 | −0.09 *** | −4.91 | −0.10 *** | −5.12 | −0.09 *** | −5.05 |
| ROE | 1.37 *** | 5.10 | 1.35 *** | 5.07 | 1.32 *** | 4.97 | 1.34 *** | 5.04 |
| SG | 0.27 *** | 3.82 | 0.29 *** | 4.08 | 0.29 *** | 4.10 | 0.29 *** | 4.04 |
| CH | −0.64 *** | −11.37 | −0.62 *** | −11.09 | −0.60 *** | −10.77 | −0.60 *** | −10.78 |
| Ownerrate | 0.54 *** | 4.71 | 0.58 *** | 5.14 | 0.56 *** | 4.97 | 0.57 *** | 5.03 |
| Fixed Effects | Year and Industry | | Year and Industry | | Year and Industry | | Year and Industry | |
| Observations | 2300 | | 2300 | | 2300 | | 2300 | |
| F-statistics | 26.40 *** | | 27.27 *** | | 27.03 *** | | 27.00 *** | |
| F-statistics of IV | - | | - | | - | | 44.21 *** | |
| Adjusted R2 | 0.332 | | 0.340 | | 0.342 | | 0.342 | |

Significance levels represented by *p*-values in the correlation analysis are as follows: *** represents a 1% significance level.

### 4.2. The Influence of De Facto CEOs on Firm Value

To empirically test Hypothesis 2, where the dependent variable is continuous and the variable of interest is a dummy variable, we conducted both OLS and 2SLS analyses. We segmented our analysis of the impacts of RealCEO and ESG activities on Tobin's Q, a proxy for firm value, into Z1, Z2, Z3, and Z4.

Z1 depicts the results of analyzing the impact of RealCEO on firm value. Z2 represents the analysis of the influence of ESG activities on firm value. Z3 illustrates the results of assessing the impact of the presence or absence of a RealCEO on firm value while controlling for ESG activities. Z4 presents the results of the 2SLS analysis using lagged variables as instruments in the Z3 model.

Table 8 presents the results of the OLS and 2SLS analyses. The empirical findings show a statistically significant negative relationship between RealCEO and firm value across Z1, Z2, Z3, and Z4. Specifically, the regression coefficient values for RealCEO in Z1, Z3, and Z4 are −0.14, −0.12, and −0.19, respectively, indicating a statistically significant negative relationship at the 1% level. This suggests that firms with RealCEOs have lower firm value compared to those without RealCEOs. To control for year and industry effects, year and industry dummy variables were incorporated. The maximum variance inflation factor (VIF) among the variables was below 2.440, indicating that there were no multicollinearity issues.

The Z2 analysis shows that the regression coefficient for ESG activities is −0.25, indicating a statistically significant negative relationship at the 1% level.

For the empirical testing of Hypothesis 2 concerning endogeneity between the dependent variable, firm value, and the independent variable, RealCEO, we used lagged variables as instrumental variables. Since there is only one instrument and one explanatory variable, the model is considered to be "just identified," and hence, we did not conduct an overidentification test. Instead, we performed a weak IV test. The F-statistic value was above 10, indicating a strong instrument, thus confirming its appropriateness as an instrument. The results from the 2SLS analysis were consistent with those obtained using OLS regression, confirming that no endogeneity concerns exist.

### 5. Conclusions

Governance is influenced by CEO type. Previous studies have reported that whether the CEO is a professional manager or owner-manager differentially impacts corporate value and financial performance [12]. However, in reality, controlling shareholders can have a significant influence on the crucial policy decisions of the company, whether or not they take office as CEO [6]. Therefore, this study empirically analyzed how corporate value and ESG activities manifest in companies where employees receive higher salaries than the CEO.

Studies have analyzed the impact of factors such as governance transparency, large corporate groups, and the influence and type of CEO on corporate value and financial performance. Research on governance transparency has used metrics such as the proportion of outside directors on the board and the presence of an independent audit committee; empirical results suggest that a higher proportion of outside directors on the board and the presence of an independent audit committee positively correlate with increased corporate value and financial performance [1,2,10]. Studies on large corporate groups have used the range of corporate groups provided by the Fair Trade Commission's corporate group portal for analysis. Empirical results suggest that the corporate value of companies belonging to large corporate groups is lower than that of companies that do not belong to such groups [12]. Research related to the influence of the CEO has analyzed the proportion of the CEO's salary to the total compensation of the management team. The empirical results suggest that corporate value decreases as the CEO's influence increases [1]. Studies concerning the type of CEO distinguish between professional managers and owner-managers based on business reports. They report that in companies with independent governance and transparent professional management, the positive effect of the compensation gap among

the CEO, executives, and corporate performance increases. Conversely, in owner-managed firms, the negative effect is mitigated [12].

In this study, we identify CEOs using compensation data, which is an objective measure of the authority and responsibilities of employees and executives. Although not representative directors, executives who received higher compensation than the representative director were defined as de facto CEOs.

Through testing Hypothesis 1, we demonstrated a statistically significant positive correlation at the 1% level between the presence of a de facto CEO and ESG activities. This finding suggests that firms with de facto CEOs are more engaged in ESG activities than those without. A de facto CEO capable of forming favorable public opinion among entities that could hold them accountable for their pursuit of personal interests, such as the media and civic groups, has an incentive to actively pursue ESG to improve their reputation and positive image. Hence, while the impact on firm value may vary depending on how ESG activities are followed, it can be concluded that firms with de facto CEOs are favorably disposed towards ESG as a means of pursuing personal benefits.

Through testing Hypothesis 2, we demonstrated a statistically significant negative correlation at the 1% level between the presence of a de facto CEO and firm value, indicating that firms with a de facto CEO have lower firm value than those without. If a company wants to enhance its value, it should increase its corporate governance transparency, ensuring that the CEO effectively holds the greatest authority and responsibility and operates under accountable management through independent policymaking bodies.

This study makes three main contributions to the literature. First, it expands the research scope related to corporate governance and ESG by identifying and analyzing de facto CEOs using compensation data, which is an objective indicator for evaluating the abilities and powers of executives. Although ESG has been actively researched in various fields worldwide, in South Korea, most studies have focused on examining the relationship between ESG and its influencing factors. This study contributes academically by expanding the research scope, examining the decision variables that affect ESG activities, and conducting an empirical analysis. This approach can be utilized in various ways in future corporate governance and ESG studies.

Second, this study empirically demonstrates that operating the board of directors, the ultimate decision-making body for corporate policy, can independently increase the transparency of governance and enhance firm value. This suggests the need to improve policymaking bodies within a company to prevent the existence of a de facto CEO who delegates legal responsibilities to representative directors but makes significant policy decisions for company management. In the long term, this can negatively affect a company's value.

Third, this study empirically illustrates that ESG activities do not always positively affect all firms. In companies with weak corporate governance, ESG activities can be used to pursue the private interests of controlling shareholders, resulting in inefficient management that reduces corporate profits and firm value. Therefore, companies must approach ESG activities cautiously and explore and implement ESG activities that suit their characteristics.

This study also has several limitations. First, the definition of our critical variable of interest, the de facto CEO, is limited because it does not consider all corporate governance relationships beyond compensation. Second, measuring ESG activities using only ESG ratings provided by the Korea ESG Standards Institute may introduce some biases.

Despite these limitations, identifying de facto CEOs using compensation, which is an objective indicator for assessing the abilities and powers of executives, and examining the impact of their existence on corporate value and ESG activities can provide insights for future research related to de facto CEOs, corporate governance, and ESG.

**Author Contributions:** Conceptualization, K.-J.B. and Y.-J.Y.; Data curation, K.-J.B.; Formal analysis, K.-J.B. and Y.-J.Y.; Investigation, K.-J.B.; Methodology, K.-J.B. and Y.-J.Y.; Project administration, K.-J.B. and Y.-J.Y.; Resources, K.-J.B.; Software, K.-J.B.; Supervision, Y.-J.Y.; Validation, K.-J.B. and Y.-J.Y.; Visualization, K.-J.B.; Writing—original draft, K.-J.B.; Writing—review and editing, Y.-J.Y. All authors have read and agreed to the published version of the manuscript.



**Funding:** This research received no external funding.

**Institutional Review Board Statement:** Not applicable.

**Informed Consent Statement:** Not applicable.

**Acknowledgments:** This paper has been written by modifying and supplementing a part of the first author's doctoral dissertation from the Department of Business Administration at Jeju National University. I would like to express my gratitude to Young-Jun Yeo for guiding my doctoral dissertation and to Gil-Hoon Kim, Soon-Yeo Jung, Chang-Youl Ko, and Jin-Soo Kim from Jeju National University for reviewing it.

**Conflicts of Interest:** The authors declare no conflict of interest.

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
