# Peer review of "The Impact of a De Facto CEO on Environmental, Social, and Governance Activities and Firm Value: Evidence from Korea"

_sustainability, doi:10.3390/su152115308_

Round 1

Reviewer 1 Report

1. The authors should elaborate the ESG activities in South Korea.

2. The importance of the types of CEO have to be mentioned in Section 1.

3. What is the research gap?

4. What is/are the thoery(ies) underpinned this study? The authors should have a brief review on those?

5. The authors have to explain the variables listed in Table 5. What is the definition? How did the authors measure those variables?

Author Response

Response to reviewer 1

Dear Reviewer:

Thank you for taking the time to review our manuscript, titled "The impact of a de facto CEO on ESG activities and firm value: Evidence from Korea," submitted to Sustainability. We are confident that the quality of our manuscript has improved as a result of incorporating your valuable suggestions. Detailed responses to your comments are provided below.

Improvement Requests1: The authors should elaborate the ESG activities in South Korea.

Response: In South Korea, the most commonly used ESG index is the one developed by the Korea ESG Standards Institute. Established in 2011, the institute annually releases ESG indices that incorporate social responsibility, governance, and environmental management. The evaluation model adheres to international standards such as the OECD Principles of Corporate Governance and ISO 26000, while also faithfully reflecting domestic laws and business environments.

This model utilizes corporate disclosure data, supervisory agency and local government records, and news media sources to collect foundational data from approximately 900 listed companies, totaling over 900 variables per company. Based on this data, an initial evaluation is conducted across 18 major categories and 265 key evaluation items, answering questions like "Is the system well-equipped to minimize ESG risks?" Additionally, an in-depth evaluation is performed, addressing 58 items such as "Has the company faced any issues that could potentially harm its value in relation to ESG?"

Upon completing the evaluation, validation procedures are undertaken to enhance the reliability of the results. Through a web-based evaluation system, bi-directional feedback is facilitated between the evaluating institution and the target companies. Final scores are then reviewed by the ESG Committee within the Korea ESG Standards Institute. Any events occurring after the evaluation period that could impact a company’s rating are flagged for review, and adjustments are made after a reexamination.

Improvement Requests2: The importance of the types of CEO have to be mentioned in Section 1.

Response: In a 2022 study by Hong and Yoo, the researchers analyzed whether corporate performance varies depending on the type of Chief Executive Officer (CEO). The study found that when the CEO is a professional manager, there is a statistically significant positive (+) relationship with corporate performance. Furthermore, the study reported that the positive (+) effects are amplified in companies with independent and transparent governance structures. Conversely, the negative (-) effects are mitigated in firms controlled by dominant shareholders. In addition to this study, previous research has indicated that the impact of the CEO type on corporate value varies depending on factors such as leadership style, expertise, personality, compensation system, and corporate strategy.

Improvement Requests3: What is the research gap?

Response: The research gap in this study is distinctive. Unlike previous studies that have used various indicators to measure the transparency of governance, this study empirically analyzes corporate governance transparency by utilizing compensation data, which is one of the objective indicators for assessing the ability and authority of employees. This approach stands in contrast to other research, which predominantly measures governance transparency through different variables. Furthermore, this study sets itself apart by empirically demonstrating that ESG (Environmental, Social, and Governance) activities in companies with weak governance could indeed impair corporate value. Thus, this research expands the scope of studies related to governance and ESG by offering new insights and contrasting perspectives. The uniqueness and contribution of this study are rooted in its approach and findings, enriching the discourse in the field of accounting and governance research.

Improvement Requests4: What is/are the thoery(ies) underpinned this study? The authors should have a brief review on those?

Response: The theoretical foundation supporting this study is the agency problem theory. According to traditional agency problem theory, agency conflicts arise in various contexts such as between employees and shareholders, and between controlling and minority shareholders. Key manifestations of agency problems include goal misalignment, information asymmetry, and resource wastage. Dominant shareholders tend to waste company resources to pursue their private benefits, and while professional managers aim to maximize shareholder value, controlling shareholders might be oriented toward the maximization of their private gains, leading to a potential misalignment of goals. Based on the theory of agency problems, prior research analyzing the impact of ESG on corporate value has reported a negative association between ESG activities and corporate value.

In this paper, we investigate the agency problems arising from the unique corporate governance structure characterized by chaebols, which are conglomerates unique to South Korea. These chaebols allow controlling shareholders to exert significant influence over corporate management with relatively small equity stakes. Specifically, the paper examines the potential for agency conflicts between controlling shareholders and professional managers appointed to represent the interests of all shareholders.

Specifically, while controlling shareholders aim for the maximization of their personal benefits, professional managers are focused on maximizing corporate value for shareholder value maximization, leading to goal misalignment. Additionally, we empirically demonstrate that controlling shareholders may misuse corporate resources for their private gains, leading to resource wastage and, consequently, a negative impact on corporate value.

Improvement Requests5: The authors have to explain the variables listed in Table 5. What is the definition? How did the authors measure those variables?

Response:

Tobin's Q: Measured as a proxy for firm value, this variable is calculated by dividing the sum of the market value of common and preferred stock and the book value of short-term and long-term debt by the total assets of the company.

TotalESG: This variable represents a firm's ESG (environmental, social, governance) activities. It is measured using the ESG Index released annually by the Korea ESG Standards Institute. A dummy variable of 1 is assigned if the firm's ESG rating is B+ or higher, and 0 otherwise.

Size: This variable represents the size of the firm. To control for the size effect, the natural logarithm of the firm's total assets is used.

LEV: This variable represents the firm's leverage ratio and is calculated by dividing the firm's total liabilities by its total equity.

CFO: This variable represents the cash flow from operating activities ratio and is measured by dividing the firm's operating cash flows by its total assets.

ROE: This variable represents the return on equity and is calculated by dividing the firm's net income by its total equity.

SG: This variable represents the firm's sales growth rate, measured by subtracting last year's sales from this year's sales and dividing by last year's sales.

BIG: This variable indicates whether the firm has been audited by one of the Big 4 accounting firms (PWC, KPMG, Deloitte, EY). It is measured as a dummy variable, where 1 indicates the firm has been audited by one of the Big 4 firms, and 0 otherwise.

CH: This variable is unique to South Korea and represents whether the firm is among the top 30 conglomerates in South Korea based on asset size (including subsidiaries). It is measured as a dummy variable, with 1 indicating a conglomerate and 0 otherwise.

Ownerrate: This variable represents the ownership percentage of the firm's largest shareholder and is measured based on the share ownership rate of the firm's single largest shareholder.

Reviewer 2 Report

The present manuscript revolves around the main idea that governance is influenced by the type of CEO. According to previous studies,  whether the CEO is a professional manager or an owner-manager strongly affects corporate value and financial performance. 

Considering that controlling shareholders can have a significant influence on the crucial policy decisions of the company, whether or not they take office as CEO. The present study empirically analyzes how corporate value and ESG activities manifest in companies where employees receive higher salaries than the CEO. 

The manuscript is well-structured, clear and well-written. There are no important issues with regard to language. The topic is interesting and relevant to the field. The article is scientifically sound. The hypotheses are well formulated, and the research method is appropriate and properly developed. All the explanations are detailed.  Figures and tables are properly presented and clearly show the data, facilitating their understanding and interpretation. The conclusions are well-founded and aligned with the arguments presented.

The references cited in the text cover a reasonable number of recent publications and are relevant.

Author Response

Response to reviewer 2

Dear reviewer:

The present manuscript revolves around the main idea that governance is influenced by the type of CEO. According to previous studies, whether the CEO is a professional manager or an owner-manager strongly affects corporate value and financial performance. 

Considering that controlling shareholders can have a significant influence on the crucial policy decisions of the company, whether or not they take office as CEO. The present study empirically analyzes how corporate value and ESG activities manifest in companies where employees receive higher salaries than the CEO. 

The manuscript is well-structured, clear and well-written. There are no important issues with regard to language. The topic is interesting and relevant to the field. The article is scientifically sound. The hypotheses are well formulated, and the research method is appropriate and properly developed. All the explanations are detailed.  Figures and tables are properly presented and clearly show the data, facilitating their understanding and interpretation. The conclusions are well-founded and aligned with the arguments presented.

The references cited in the text cover a reasonable number of recent publications and are relevant.

Response: Thank you for your positive comments.

Reviewer 3 Report

Dear Authors,

I am pleased to have the opportunity to review your manuscript, titled "The impact of a de facto CEO on ESG activities and firm value: Evidence from Korea," submitted to Sustainability. Your study delves into a compelling topic by examining the impact of "substantial CEOs," who are the highest paid in their companies, on corporate governance. It finds that such CEOs are more engaged in ESG activities, possibly to improve public image while concealing self-serving behaviors. The study also reveals that these CEOs contribute to lower corporate value, likely due to prioritizing personal interests over long-term profitability. To fully exploit the study's potential, there are some issues which need to be addressed or at least considered in your manuscript. I hope these comments can help you improve the manuscript.

The authors might enhance the structure of their research by developing a comprehensive framework or overarching theoretical model to justify the combined focus on ESG activities and firm value within a single research paper. In the current form, a perception of fragmentation arises as ESG activities and firm value are discussed independently.

Regarding the methodology section, I have several significant concerns:

·         My primary concern revolves around your chosen ESG measurement. ESG is typically measured as a continuous variable (for example, scores from Sustainalytics, CSRHub, or Bloomberg) or as aggregated count variables (such as scores from ASSET 4 or KLD). Your use of a grade-based threshold (B+ or above) to measure ESG is unconventional and needs comprehensive justification. You should corroborate this approach with reference to existing literature or through robust analysis where different thresholds are tested for consistency in the results. By doing so, you can derive more verifiable outcomes and defend your research approach more effectively. Ensuring your ESG measurement is robust and justifiable is key to grounding your findings in accepted methodologies.

·         Equally, there seems to be a lack of supporting literature to justify your measurement of ‘de facto CEO’. The term ‘representative directors’ is used ambiguously without a comprehensive explanation. An explanation or definition in the context of your study, supported by appropriate references, would enhance the understanding of the reader. Moreover, it is critical to ensure that the concept aligns with recognized practices, theories, and definitions in the existing body of literature to promote the robustness and credibility of your research. Therefore, elaborating on these ideas within the context of your study is strongly recommended.

·         Your inclusion of control variables also warrants substantiation from existing literature. It’s crucial to explain the reason behind choosing each control variable and how it is expected to affect the results. This can be best achieved by citing relevant prior studies that used similar control variables, thus solidifying the research foundation of your analysis. Perhaps, offering a theoretical or empirical justification for the selection of these variables could make your research methodology more transparent, accurate, and easy to comprehend for your readers.

·         The paper utilizes Tobin’s Q as a measure of firm value, but doesn’t adequately justify this choice, especially considering the array of other potential measures. It’s essential to explain why Tobin’s Q was selected over other common metrics, such as Price/Earnings Ratio (P/E), or Price/Sales Ratio (PSR). Ensure you provide a theoretical or empirical rationale, referencing relevant literature, for your choice, showing how it best serves your specific research objectives, and adds value to your study. A well-backed justification will validate the appropriacy and effectiveness of your chosen measurement.

·         While writing Equations 1 and 2, I strongly recommend using an equation function available in applications like Microsoft Word, LaTeX, or Mathpix. It appears there are several errors in the current equations. Notably, in Equation 1, the subscript of the coefficient isn’t correctly written; it should be in lowercase rather than uppercase. Utilizing an equation editor should help prevent such errors, improving the accuracy and readers’ comprehension of your mathematical representations.

·         In relation to Table 6, it would also be beneficial to include the corresponding p-values for each correlation coefficient.

·         The choice to report the Wald test values in your regression results is somewhat unconventional. Generally, in regression output, standard errors or t-values are presented as they are more informative about the extent and significance of the deviation from the estimated coefficients. I recommend adopting this more typical approach in your regression analysis, unless a strong rationale for using Wald test values is provided and justified in your study.

·         One major concern involves the lack of consideration for endogeneity issues. A discussion on your approach to address this issue is noticeably absent. For example, considering the use of a two-stage least squares (2SLS) method instead of ordinary least squares (OLS) could potentially better manage endogeneity concerns. Alternatively, it might be beneficial to include a one-year lag of the independent variables to mitigate the potential issue of reverse causality.

To summarize, I believe you addressed important gaps literature, and I hope that the authors will be able to use my feedback to develop it further. Good luck! 

N/A

Author Response

Response to reviewer 3

Dear Reviewer:

Thank you for taking the time to review our manuscript, titled "The impact of a de facto CEO on ESG activities and firm value: Evidence from Korea," submitted to Sustainability. We are confident that the quality of our manuscript has improved as a result of incorporating your valuable suggestions. Detailed responses to your comments are provided below.

Improvement Requests1:

ESG measurement. ESG is typically measured as a continuous variable (for example, scores from Sustainalytics, CSRHub, or Bloomberg) or as aggregated count variables (such as scores from ASSET 4 or KLD). Your use of a grade-based threshold (B+ or above) to measure ESG is unconventional and needs comprehensive justification. You should corroborate this approach with reference to existing literature or through robust analysis where different thresholds are tested for consistency in the results. By doing so, you can derive more verifiable outcomes and defend your research approach more effectively. Ensuring your ESG measurement is robust and justifiable is key to grounding your findings in accepted methodologies.

Response: This paper conducts empirical analysis specifically targeting the South Korean market; therefore, it utilizes the ESG index provided by the Korea ESG Standards Institute, which is the most commonly used in South Korea. The ESG evaluation model from the Korea ESG Standards Institute not only adheres to international standards such as OECD Principles of Corporate Governance and ISO 26000, but it is also designed to faithfully reflect domestic laws and business environments. This model collects over 900 pieces of foundational data for each company from approximately 900 listed companies by utilizing corporate disclosure documents, data from supervisory agencies, local government institutions, and media sources. It then undergoes a comprehensive evaluation based on 18 main categories and 265 key evaluation items, along with 58 advanced evaluations.

Since the proportional odds assumption holds, the results from the ordinal logistic analysis targeting all grades (8 grades) were not significantly different from the binary logistic analysis. Therefore, we can conclude that the ESG dummy variable is robust.

Improvement Requests2: there seems to be a lack of supporting literature to justify your measurement of ‘de facto CEO’. The term ‘representative directors’ is used ambiguously without a comprehensive explanation. An explanation or definition in the context of your study, supported by appropriate references, would enhance the understanding of the reader. Moreover, it is critical to ensure that the concept aligns with recognized practices, theories, and definitions in the existing body of literature to promote the robustness and credibility of your research. Therefore, elaborating on these ideas within the context of your study is strongly recommended.

Response: In preceding studies related to corporate governance, the types of CEOs have been analyzed by categorizing them into professional managers and    owner-managers. Ahn and Seo[7] were the first to define those shareholders who, within the owner-managers, evade legal responsibility while exercising influence on the company as “de facto CEOs,” measured using compensation data. The measurement of CEO influence is inspired by the CPS (CEO pay slice) proposed by Bebchuk et al. [15], utilizing compensation data for measurement. The CPS denotes the ratio of the CEO's salary to the total compensation of the top five executives within a company. Since Bebchuk et al. [15] first presented CPS as a metric to measure the influence of CEOs on corporate control and analyzed its relation with corporate value, it has been used widely.

This study followed the idea that, based on agency theory, there is a negative correlation with corporate value due to the agency costs of goal incongruity and resource wastage, in companies where professional managers represent shareholders and dominant shareholders exercise substantial influence on corporate control with minority interests.

Improvement Requests 3: Your inclusion of control variables also warrants substantiation from existing literature. It’s crucial to explain the reason behind choosing each control variable and how it is expected to affect the results. This can be best achieved by citing relevant prior studies that used similar control variables, thus solidifying the research foundation of your analysis. Perhaps, offering a theoretical or empirical justification for the selection of these variables could make your research methodology more transparent, accurate, and easy to comprehend for your readers.

Response: In Model 1, we used firm size (SIZE), leverage ratio (LEV), return on equity (ROE), sales growth rate (SG), affiliation to large business groups (CH), foreign ownership ratio (Foreign), and annual and industry dummy variables as control variables that influence the TotalESG score. We included SIZE as a control variable because larger firms tend to have lower unsystematic risk and a more robust financial structure, enabling them to focus not only on their core profit-generating business but also on non-profit activities like ESG. The inclusion of ROE, a profitability indicator, suggests that companies with higher profitability can invest more resources in ESG activities. The SG metric represents the firm's growth prospects, which can also lead to higher expected profitability in the future and, therefore, more resources allocated to ESG initiatives. The CH variable was added because large business groups are more likely to be socially responsible and hence more involved in ESG activities. A higher foreign ownership ratio (Foreign) indicates that the firm may be subject to more stringent global ESG standards, justifying its inclusion as a control variable. To control for year-to-year and industry-specific influences, we included annual and industry dummy variables.

In Model 2, Tobin’s Q, calculated as the market value of equity plus the book value of debt divided by total assets, was used as a proxy for firm value. The control variables for Tobin’s Q included TotalESG, SIZE, LEV, ROE, SG, CH, ownership by the largest shareholder (OwnerRate), and annual and industry dummy variables. SIZE and LEV were included to control for the influence of risk on firm value, given that larger firms or those with lower leverage are less susceptible to bankruptcy risks. ROE was included to control for the effects of profitability on firm value. SG, a growth indicator, was included since a higher growth rate can influence firm value. The cash flow from operations ratio (CFO) was also included as firms with a higher proportion of cash in their earnings are generally considered to have higher-quality earnings, positively affecting firm value. OwnerRate was included because it can have both advantages and disadvantages concerning timely and unilateral decision-making, impacting firm value. CH, a variable indicating affiliation to large business groups, was added from a governance perspective. To control for annual and industry-specific effects, annual and industry dummy variables were incorporated into the model.

Improvement Requests4: The paper utilizes Tobin’s Q as a measure of firm value, but doesn’t adequately justify this choice, especially considering the array of other potential measures. It’s essential to explain why Tobin’s Q was selected over other common metrics, such as Price/Earnings Ratio (P/E), or Price/Sales Ratio (PSR). Ensure you provide a theoretical or empirical rationale, referencing relevant literature, for your choice, showing how it best serves your specific research objectives, and adds value to your study. A well-backed justification will validate the appropriacy and effectiveness of your chosen measurement.

Response: Previous studies have used Tobin's Q as a proxy for firm value. Specifically, Dahya et al. (2008) reported a positive relationship between firm performance (measured as Tobin's Q) and board independence. Loderer and Martin (1997) examined the relationship between insider ownership and performance (measured as Tobin’s Q and 6-days’ CAR: Cumulative Abnormal Returns) for 867 U.S. listed firms that underwent mergers between 1978 and 1988. Black and Kim (2012) also analyzed the relationship between corporate governance and firm value, finding that higher levels of corporate governance are associated with higher Tobin's Q values. According to the results from 2SLS and 3SLS analyses, these findings were consistent with those from OLS estimations but with even greater statistical significance. In addition to these, several other studies have also defined Tobin's Q as a proxy for firm value in their analyses (Block 1999; Lin et al. 2003; Bethel et al. 1998).

Improvement Requests5: While writing Equations 1 and 2, I strongly recommend using an equation function available in applications like Microsoft Word, LaTeX, or Mathpix. It appears there are several errors in the current equations. Notably, in Equation 1, the subscript of the coefficient isn’t correctly written; it should be in lowercase rather than uppercase. Utilizing an equation editor should help prevent such errors, improving the accuracy and readers’ comprehension of your mathematical representations.

Response: In the paper, the subscripts in Model 1 have been updated to lowercase.

Improvement Requests6: Table 6, it would also be beneficial to include the corresponding p-values for each correlation coefficient.

Response: The p-values have been successfully incorporated into the results of the correlation analysis in Table 6.

Improvement Requests7: The choice to report the Wald test values in your regression results is somewhat unconventional. Generally, in regression output, standard errors or t-values are presented as they are more informative about the extent and significance of the deviation from the estimated coefficients. I recommend adopting this more typical approach in your regression analysis, unless a strong rationale for using Wald test values is provided and justified in your study.

Response: The Wald test values have been updated to regression analysis values.

Improvement Requests8: One major concern involves the lack of consideration for endogeneity issues. A discussion on your approach to address this issue is noticeably absent. For example, considering the use of a two-stage least squares (2SLS) method instead of ordinary least squares (OLS) could potentially better manage endogeneity concerns. Alternatively, it might be beneficial to include a one-year lag of the independent variables to mitigate the potential issue of reverse causality.

Response: To test for endogeneity between the dependent variable, ESG, and the independent variable, RealCEO, in Model 1, we utilized the proportion of treasury shares owned by the company as an instrument variable, following the method proposed by Park Jin-hyuk and Lee Jang-woo (2022). Companies with a RealCEO can use treasury shares as a means to defend and maintain managerial control; therefore, it can be interpreted that the more treasury shares a company holds, the more concern it has for the maintenance and defense of its managerial control. For this reason, the proportion of treasury shares in a company has a direct correlation with the existence of a RealCEO but is considered to have no direct impact on ESG evaluations.

In Model 2, lagged variables were used as instrument variables to test for endogeneity between the dependent variable, firm value, and the independent variable, RealCEO.

Using these instrument variables in each model for 2SLS (two-stage least squares) analysis, the results were not significantly different from those obtained using OLS (ordinary least squares), confirming that endogeneity is not an issue.

I have attached the results of the robustness analysis for ESG variables in the form of rank-ordered logistic regression results in an Excel file. For the robustness analysis of the empirical results, I have attached the 2SLS analysis results in an Excel file.

Round 2

Reviewer 1 Report

-

 Moderate editing of English language required

Author Response

Response to reviewer 1

Dear Reviewer,

Thank you for taking the time to review our manuscript, titled “The impact of a de facto CEO on ESG activities and firm value: Evidence from Korea,” submitted to Sustainability. We are confident that the quality of our manuscript has improved as a result of incorporating your valuable suggestions. Detailed responses to your comments are provided below.

Improvement Requests1: Moderate editing of English language required.

Response: I have completed the proofreading through a professional English editing service. I will submit the English Proofreading Certificate along with the manuscript.

Please adjust as per the formalities or specifics of your communication context.

Reviewer 3 Report

1.       It would enhance the clarity and rigour of the discussion if supporting references were cited when discussing the inclusion of each control variable.

2.       Equation (1) appears to be incorrectly formatted. Specifically, the indices (i.e., 1, 2, 3, ...) alongside α should be in lowercase to maintain consistency with Equation (2). While Equation (2) seems to be correct, it is not formatted professionally for an academic paper. It is advisable to utilize the equation function in Microsoft Word or third-party software for neat and accurate formatting of equations. I would recommend using Mathpix for this purpose.

3.       In Table 4, the names of the first two variables appear to be incorrect. Kindly rectify these to maintain accuracy and consistency throughout the document.

4.       The authors mention the use of instrumental variables, however, the choice of these variables remains unclear. The authors offer some theoretical justifications, which I am fine with. I noticed these justifications are only included in the response letter. Furthermore, I observed a lack of analysis regarding the weak Instrumental Variable (IV) test and the Overidentification test. It would be beneficial to include these tests to ensure the robustness and validity of the instrumental variables used in the study.

Author Response

Point by point response is added in a seperate folder below. Kindly refer to the attached file "Response to Reviewer 3". 

Round 3

Reviewer 3 Report

I recommend "accept".